# Spatial and Temporal Hierarchy for Autonomous Navigation Using Active Inference in Minigrid Environment

**Daria de Tinguy** [1,*] [ID], **Toon Van de Maele** [2], **Tim Verbelen** [2,*] and **Bart Dhoedt** [1]

1    IMEC, Ghent University, 9000 Gent, Belgium; bart.dhoedt@ugent.be
2    VERSES AI Research Lab, Los Angeles, CA 90016, USA; toon.vandemaele@verses.ai
*    Correspondence: daria.detinguy@ugent.be (D.d.T.); tim.verbelen@verses.ai (T.V.)

**Abstract:** Robust evidence suggests that humans explore their environment using a combination of topological landmarks and coarse-grained path integration. This approach relies on identifiable environmental features (topological landmarks) in tandem with estimations of distance and direction (coarse-grained path integration) to construct cognitive maps of the surroundings. This cognitive map is believed to exhibit a hierarchical structure, allowing efficient planning when solving complex navigation tasks. Inspired by human behaviour, this paper presents a scalable hierarchical active inference model for autonomous navigation, exploration, and goal-oriented behaviour. The model uses visual observation and motion perception to combine curiosity-driven exploration with goal-oriented behaviour. Motion is planned using different levels of reasoning, i.e., from context to place to motion. This allows for efficient navigation in new spaces and rapid progress toward a target. By incorporating these human navigational strategies and their hierarchical representation of the environment, this model proposes a new solution for autonomous navigation and exploration. The approach is validated through simulations in a mini-grid environment.

**Keywords:** active inference; autonomous navigation; spatial hierarchy; temporal hierarchy; predictive coding

## 1. Introduction

The development of autonomous systems that can navigate in their environment is a crucial step towards building intelligent agents that can interact with the real world. Just as animals possess the ability to navigate their surroundings, developing navigation skills in artificial agents has been a topic of great interest in the field of robotics and artificial intelligence [1–3]. This has led to the exploration of various approaches, including taking inspiration from animal navigation strategies (e.g., building cognitive maps [4]), as well as state-of-the-art techniques using neural networks [5]. However, despite significant advancements, there are still limitations in both non-neural-network- and neural-network-based navigation approaches [2,3].

In the animal kingdom, cognitive mapping plays a crucial role in navigation. Cognitive maps allow animals to understand the spatial layout of their surroundings [6–8], remember key locations, solve ambiguities from context [9], and plan efficient routes [9,10]. By leveraging cognitive mapping strategies, animals can successfully navigate complex environments, adapt to changes, and return to previously visited places.

In the field of robotics, traditional approaches have been explored to develop navigation systems. These approaches often rely on explicit mapping and planning techniques, such as grid-based [11,12] and/or topological maps [13,14], to guide agent movement. While these methods have shown some success, they suffer from limitations in handling complex spatial relationships and dynamic environments as well as scalability issues as the environment grows larger [2,3,15].

To overcome the limitations of these non-neural network approaches, recent advancements have focused on utilising neural networks for navigation [5,16–18]. Neural-network-based models, trained on large datasets, have shown promise in learning navigational policies directly from raw sensory input. These models can capture complex spatial relationships and make decisions based on learned representations. However, the current neural-network-based navigation approaches also face challenges, including the need for extensive training data, limitations in generalisation to unseen environments, distinguishing aliased areas, and the difficulty of handling dynamic and changing environments [2].

To address these challenges, we propose building world models based on active inference. Active inference is a framework allowing agents to actively gather information through perception, select and execute actions in their environment, and learn from accumulated experiences [19,20]. World models, within this framework, form internal representations of the world. Agents endowed with a world model and engaged in active exploration continually update their internal understanding of the environment, empowering them to make well-informed decisions and predictions [21,22]. This principled approach enables continuous belief updates and active information gathering, facilitating effective navigation [20].

Noting that biological agents are building hierarchically structured models, we construct multi-level world models as hierarchical active inference. Hierarchical active inference warrants agents to utilise layers of world models, facilitating a higher level of spatial abstraction and temporal coarse-graining. It enables learning complex relationships in the environment and allows more efficient decision-making processes and robust navigation capabilities [23]. By incorporating hierarchical structures into active inference-based navigation systems, agents can effectively handle complex environments and perform tasks with greater adaptability [24].

In this paper, in order to improve the agent's ability to navigate autonomously and intelligently, we propose a hierarchical active inference model composed of three layers. Our proposed system's highest layer is able to learn the environment structure, remember the relationship between places, and navigate without prior training in a familiar yet new world. The second layer, the allocentric model, learns to predict the local structure of rooms, while the lowest level, our egocentric model, considers the dynamic limitations of the environment. We aim to enhance the agent's ability to navigate through complex and dynamic environments while maintaining scalability and adaptability.

Our contributions can be summarised as follows:

- We present a system combining hierarchical active inference with world modelling for task-agnostic autonomous navigation.
- Our system uses pixel-based visual observations, which show promise for real-world scenarios.
- Our model learns the structure of the environment and its dynamic limitations in order to form an internal map of the full environment independently of its size, without requiring more computation as the environment scales up.
- Our system can plan long-term without worrying about look-ahead limitations.
- We evaluate the system in a mini-grid room maze environment [25], showing the efficiency of our method for exploration and goal-related tasks, compared against other reinforcement learning (RL) models and other baselines.
- We quantitatively and qualitatively assess our work, showing how our hierarchical active inference world model fares in accomplishing given tasks, how it resists aliasing, and how it learns the structure of the environment.

The subsequent sections of this paper delve into the details of our proposed approach, including the theoretical foundations of active inference and hierarchical active inference, the architecture of our navigation system, experimental results, and a comprehensive discussion of the advantages and limitations of our approach.

## 2. Related Work

Navigating complex environments is a fundamental challenge for both humans and artificial agents. To solve navigation, traditional approaches often address simultaneous localisation and mapping (SLAM) by building a metric (grid) map [11,12] and/or topological map of the environment [13,14]. Although there is progress in this area, Placed et al. [3] state that active SLAM may still fail to be fully autonomous in complex environments. The current approaches are also still lacking in distinct capabilities important for navigation, such as predicting the uncertainty over robot location, abstracting over features of the environment (e.g., having a semantic map instead of a precise 3D map), and reasoning in dynamic, changing spaces. The recent studies have explored the adoption of machine learning techniques to add autonomy and adaptive skills in order to learn how to handle new scenarios in real-world situations. Reinforcement learning (RL) typically relies on rewards to stimulate agents to navigate and explore. In contrast, our model breaks away from this convention, as it does not necessitate the explicit definition of a reward during agent training. Moreover, despite the success of recent machine learning, these techniques typically require a considerable amount of training data to build accurate environment models. This training data can be obtained from simulation [26,27]; provided by humans (either by labelling, as in the works in [28,29] or by demonstration, as in [30]); or by gathering data in an experimental setting [16,31,32]. These methods all aim to predict the consequences of actions in the environment but typically generalise poorly across environments. As such, they require considerable human intervention when deployed in new settings [2]. We aim to reduce both the human intervention and the quantity of data required for training by simultaneously familiarising the agent with the structure and dynamics found in its environment.

When designing an autonomous adaptable system, nature is a source of inspiration. Tolman's cognitive map theory [33] proposes that brains build a unified representation of the spatial environment to support memory and guide future actions. More recent studies postulate that humans create mental representations of spatial layouts to navigate [6], integrating routes and landmarks into cognitive maps [7]. Additionally, the research into neural mechanisms suggests that spatial memory is constructed in map-like representations fragmented into sub-maps with local reference frames [34]; meanwhile, hierarchical planning is processed in the human brain during navigation tasks [9]. The studies of Balaguer et al. [9] and Tomov et al. [10] show that hierarchical representations are essential for efficient planning for solving navigation tasks. Hierarchies provide a structured approach for agents to learn complex environments, breaking down planning into manageable levels of abstraction and enhancing navigation capabilities, both spatially (sub-maps) and temporally (time-scales). Thus, our model incorporates these elements as the foundation of its operation.

The concept of hierarchical models has gained interest in navigation research [13,24]. Hierarchical structures enable agents to learn complex relationships within the environment, leading to more efficient decision-making and enhancing adaptability in dynamic scenarios. There are two main types of hierarchy, both considered in our work: temporal—planning over a sequence of timesteps [35–38]—and spatial—planning over structures [13,23,39,40].

In order to navigate without teaching the agent how to do so, we use the principled approach of active inference (AIF), a framework combining perception, action, and learning. It is a promising avenue for autonomous navigation [22]. By actively exploring the environment and formulating beliefs, agents can make informed decisions. Within this framework, world models play a pivotal role in creating internal representations of the environment and facilitating decision-making processes. A few models have proposed combining AIF and hierarchical models for navigation. Safron et al. [41] proposes a hierarchical model composed of two layers of complexity to learn the structure of the environment. The lowest level infers the state of each step while the higher level represents locations, created in a more coarse manner. Large, complex, aliased, and/or dynamic environments are challenges to this model. Nozari et al. [42] construct a hierarchical system by using a

dynamic Bayesian network (DBN) over a naive and an expert agent, in which the naive agent learns temporal relationships, with the highest level capturing semantic information about the environment and low-level distributions capturing rough sensory information with their respective evolution through time. This system, however, requires expert data to be trained by imitation learning, which limits the performance of the model to that of the expert. Our study focuses on familiarising the model with environmental structures rather than learning optimal policies within environments. This approach enhances the model's autonomy and adaptability to dynamic changes. Furthermore, the incorporation of spatial and temporal hierarchical abstractions effectively mitigates aliasing ambiguity and extends the agent's planning horizon for improved decision-making.

Collectively, these studies provide insights into the cognitive mapping strategies used by humans, the benefits of hierarchical representations in navigation, and the application of active inference and world models to afford decision-making in the environment. The concept of hierarchical active inference offers a possible foundation for achieving robust and efficient navigation through complex and dynamic environments. Following this line of thinking, our work proposes a new alternative to navigate in environments using pixel-based hierarchical generative models to learn the world and active inference to navigate through it.

## 3. Methods

This section presents a breakdown of the navigation framework proposed in this work. It is divided into several subsections, starting with an exploration of world models and their importance in capturing the environment. We then delve into active inference, planning through inference, and our hierarchical active inference model. Next, we discuss the specific components of our model, including the egocentric model, allocentric model, and cognitive map. The subsection on navigation covers key mechanisms such as curiosity-driven exploration, uncertainty resolution, and goal-reaching. Finally, we conclude with a brief overview of the training process.

### 3.1. World Model

We will first introduce the concept of world models in the context of navigation. Any agent, artificial or natural, can only sense its surroundings through sensory observations and change its surroundings through actions. This concept of a statistical boundary, known as a Markov blanket, plays a crucial role in defining the information flow between an agent and its environment [20,43].

The agent's world model can be defined as partially observable, corresponding to a partially observable Markov decision process (POMDP). In the framework of active inference, those world models are generative: they capture how hidden causes generate observations through actions. Given a set of observations $o$ and actions $a$, the agent creates a latent state $s$, representing its belief about the world. This corresponds to the probability distribution $P(\tilde{s}|\tilde{o}, \tilde{a}, \pi)$, where tildes are used to denote sequences, defining the agent's belief states, observations, actions, and policies. In this formalism, a policy $\pi$ is nothing more than a series of actions $a_{t:T}$ from time $t$ up until some horizon $T$.

We assume the world model is Markovian without loss of generality, so that the agent's state $s_t$ at time-step $t$ is only influenced by the prior state $s_{t-1}$ and action $a_{t-1}$.

Technically, the generative model is factorised as follows, using the notation explained above [37]:

$$P(\tilde{s}, \tilde{o}, \tilde{a}, \pi) = P(s_0)P(\pi)\prod_{t=1}^{T} P(o_t|s_t)P(s_t|s_{t-1}, a_{t-1})P(a_{t-1}|\pi) \tag{1}$$

### 3.2. Active Inference

The Markov blanket acts as a barrier between the agent and the environment, restricting the agent's direct knowledge of the world's state. Consequently, the agent must rely on observations to gauge the effects of its actions. This necessitates Bayesian inferences to revise beliefs about potential state values, based on observed actions and their corresponding observations. In fact, the agent uses the posterior belief $P(\tilde{s}|\tilde{o}, \tilde{a})$ to infer its belief state $s$ [19].

In practice, calculating the true posterior in this form, derived purely from the Bayes rule, is usually intractable directly from the given joint model in Equation (1).

To overcome this problem, the agent employs variational inference and approximates the true posterior by some approximate posterior distribution $Q(\tilde{s}|\tilde{o}, \tilde{a})$, which is in a tractable form [44].

The estimated posterior distribution can be decomposed as the model proposed in (1):

$$Q(\tilde{s}|\tilde{o}, \tilde{a}) = Q(s_0|o_0) \prod_{t=1}^{T} Q(s_t|s_{t-1}, a_{t-1}, o_t) \qquad (2)$$

This approximate posterior maps from observations and actions to internal states used to reason about the world. We assume that the agent has access to a perfect proprioceptive feedback channel, implying that all executed actions are fully observed. Therefore, the policy $\pi$ does not appear in Equation (2). For future time-steps, this assumption does not hold, since the agent needs to infer the policy $\pi$ to select actions from.

The agent is assumed to act according to the free energy principle, which states that all agents aim to minimise their variational free energy [19]. Given our generative model, we can formalise the variational free energy $F$ in the following way [37]:

$$\begin{aligned} F &= \mathbb{E}_{Q(\tilde{s}|\tilde{a}, \tilde{o})}[log Q(\tilde{s}|\tilde{a}, \tilde{o}) - log P(\tilde{s}, \tilde{a}, \tilde{o})] \\ &= \underbrace{D_{KL}[Q(\tilde{s}|\tilde{a}, \tilde{o})||P(\tilde{s}|\tilde{a}, \tilde{o})]}_{\text{posterior approximation}} - \underbrace{log P(\tilde{o})}_{\text{log evidence}} \\ &= \underbrace{D_{KL}[Q(\tilde{s}|\tilde{a}, \tilde{o})||P(\tilde{s}, \tilde{a})]}_{\text{complexity}} - \underbrace{\mathbb{E}_{Q(\tilde{s}|\tilde{a}, \tilde{o})}[log P(\tilde{o}|\tilde{s})]}_{\text{accuracy}} \end{aligned} \qquad (3)$$

This equation describes the perception process over past and present observations, wherein minimising the variational free energy leads to the approximate posterior becoming increasingly aligned with the true posterior beliefs. Essentially, this means that the process involves forming beliefs about hidden states that offer a precise and concise explanation of observed outcomes while minimising complexity. Complexity, in this case, is the difference between prior and posterior beliefs, indicating how much one adjusts their belief when moving from prior to posterior [39].

### 3.3. Planning as Inference

In active inference, agents are expected to take actions to minimise the free energy in the future. Minimising free energy with respect to future observations encourages the agent to obtain additional observations in order to maximise its evidence and can, thus, be employed as a natural strategy for exploration. However, as future observations and actions are not available to the agent, the agent minimises its expected free energy (EFE). To calculate this expected free energy G, the effect of adopting several policies (i.e., sequences of actions) on the future free energy is analysed.

$$G(\pi) = \sum_{\tau'} G(\pi, \tau') \qquad (4)$$

The expected free energy $G(\pi, \tau')$ for a certain policy $\pi$ and time-step $\tau'$ in the future for the generative model is defined as

$$G(\pi, \tau') = \underbrace{\mathbb{E}_{Q(o_{\tau'}, s_{\tau'}|\pi)}[ln(Q(s_{\tau'}|\pi)) - ln(Q(s_{\tau'}|o_{\tau'}, \pi))]}_{\text{information gain term}} - \underbrace{\mathbb{E}_{Q(o_{\tau'}, s_{\tau'}|\pi)}[ln(P(o_{\tau'}))]}_{\text{utility term}} \quad (5)$$

The expected free energy naturally balances the agent's drive towards its preferences, i.e., utility value, with the expected uncertainty of the path towards the goal, i.e., information gain [24,45].

To navigate effectively using active inference, the agent considers the current knowledge about the environment and selects policies taking into account the expected surprise at future time-steps [46].

While the dependency on policies in the prior over states can be omitted, the agent's desire to attain its preferred world states remains evident, regardless of which policy it pursues. The expected free energy is calculated for each future timestep the agent considers and is then aggregated to infer the most likely sequence of actions to reach a preferred state. This belief in policies is achieved through

$$P(\pi) = \sigma(-\gamma G(\pi)) \quad (6)$$

Where $\sigma$ the softmax function is tempered with a temperature parameter $\gamma$, converting the expected free energy of policies into a categorical distribution over policies. By using active inference, planning is transformed into an inference problem, with beliefs about policies proportionate to their expected free energy. The softmax temperature $\gamma$ represents the agent's confidence in its current beliefs over policies. Overall, this inference allows the agent to plan ahead and optimise its behaviour over time, taking into account the uncertainty and complexity of the environment to achieve its goals. This is necessary for high-level cognitive processes such as reasoning, planning, and decision-making [24,46].

### 3.4. A Hierarchical Active Inference Model

Active inference enables us to plan across a span of time; however, employing a non-hierarchical model that captures the environment in a single state or layer exhibits numerous limitations. Such models are often weak to aliasing as they lack sufficient abstraction to distinguish identical observations. Second, they possess a short-term memory and a necessarily limited horizon to infer policies, a measure taken to avoid intractable calculations. Those two elements render long-term planning a challenging endeavour. Moreover, those models often lack adaptability in case of unexpected changes in the environment. Finally, the larger the environment is, the more computational resources might be required for such a model to form a full comprehensive representation [45,47].

Therefore, in navigation, hierarchical models are sought after for resulting gains in abstraction, generalisation, and adaptability by adding levels to capture hierarchical structures and relationships [24,48].

Based on these motivations, we propose a hierarchical generative model consisting of three layers of reasoning functioning at nested timescales, aiming for more flexible reasoning over time and space (see Figure 1). In order of a decreasing level of abstraction, the layers of the model are: (a) the cognitive map, creating a coherent topological map; (b) the allocentric model, representing space; and (c) the egocentric model, modelling motions. The structure of the environment is inferred over time by agglomerating visual observations into representations of distinct places (e.g., rooms), while the highest level discovers the connectivity structure of the maze as a graph. The notations used from this point onward in this paper are described in Table 1. The full joint distribution of the generative model can be written down as in Equation (7), where we explicitly index the three distinct nested timescales with $T$, $t$, and $\tau$, respectively:

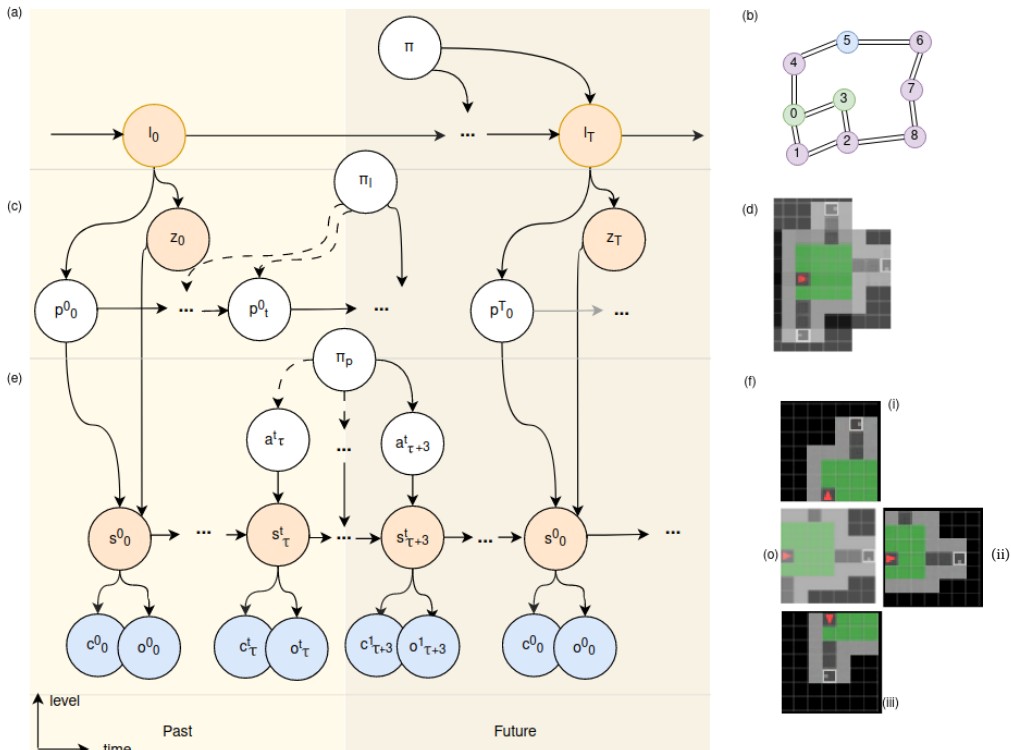

**Figure 1.** Our generative model unrolled in time and levels as defined in Equation (7). The left figure shows the graphical model of the 3-layer hierarchical active inference model consisting of (**a**) the cognitive map, (**b**) the allocentric model, and (**c**) the egocentric model, each operating at a different time scale. The orange circles represent latent states that have to be inferred, the blue circles denote observable outcomes, and the white circles are internal variables to be inferred. The right part visualises the representation at each layer. The cognitive map is represented as (**d**) a topological graph composed of all the locations (*l*) and their connections, in which each location is stored in a distinct node. The allocentric model (**e**) infers place representations (*z*) by integrating sequences of state (*s*) and poses (*p*), from which the room structure can be generated. The egocentric model (**f**) imagines future observations given the current position, state (*s*), and possible actions (*a*). Here, (**o**) depicts an actual observation (*o*), and the predicted observations of the possible actions turn left (**i**), move forward (**ii**), and turn right (**iii**).

$$P(\tilde{o}, \tilde{z}, \tilde{s}, \tilde{l}, \pi, \tilde{\pi}_l, \tilde{\pi}_p) = P(\pi) \prod_T P(z_T, p_T^0 | l_T) P(l_T | \pi) P(\pi_l)$$
$$\prod_t P(s_0^t | z_T, p_t^T) P(p_t^T | \pi_l, p_0^T) P(\pi_p) \tag{7}$$
$$\prod_\tau P(s_{\tau+1}^t | s_\tau^t, a_T^t) P(a_\tau^t | \pi_p) P(o_\tau^t, c_\tau^t | s_\tau^t)$$

At the top layer of the generative model, we see the **cognitive map**, as depicted in Figure 1a, which operates at the coarsest time scale ($T$). Each tick at this time scale corresponds to a distinct location ($l_T$), integrating the initial positions ($p_0^T$) of the place ($z^T$). These locations are represented as nodes in a topological graph, as shown in Figure 1d. As the agent moves from one location to another, edges are added between nodes, effectively learning the structure of the maze. To maintain the spatial relationship between locations, the agent utilises a continuous attractor network (CAN), similar to [49], keeping track of its relative rotation and translation. As a result, the cognitive map forms a comprehensive representation of the environment, enabling the agent to navigate and gain an understanding of its surroundings.

**Table 1.** Description of the variables used in our model.

| Notation | Associated Meaning |
|:---:|:---:|
| $l$ | location, experience |
| $z$ | place, room, allocentric state |
| $p$ | pose, position |
| $s$ | egocentric state |
| $a$ | action |
| $o$ | observation |
| $c$ | collision |
| $\pi_x$ | policy, sequence of x |

The middle layer, the **allocentric model**, depicted in Figure 1b, plays a vital role in building a coherent representation of the environment, referred to as $z_T$. This model operates at a finer time scale ($t$), generating a belief about the place by integrating a sequence of observations ($s_{0:t}^T$) and poses ($p_{0:t}^T$) to create this representation [50,51]. The resulting place, as shown in Figure 1d, defines the environment based on accumulated observations. When the agent transitions from one place to another and the current observations no longer align with the previously formed prediction of the place, the allocentric model resets its place description and gathers new evidence to construct a representation of the newly discovered room ($z_{T+1}$). This advancement corresponds to one tick on the coarser time scale, and the mid-level time scale $t$ is reset to 0.

The lowest layer is called the **egocentric model**, shown in Figure 1c, which operates at the finest time scale ($\tau$). This model utilises the prior state ($s_\tau^t$) and current action ($a_{\tau+1}^t$) to infer the current observation ($o_{\tau+1}^t$) [37]. By considering its current position, the model generates potential future trajectories while incorporating environmental constraints, such as the inability to pass through walls. Figure 1f showcases the current observation at the centre (o) and visualises the imagined potential observations if the agent were to turn left (i), right (iii), or move forward (ii).

It is important to observe that these three levels operate at different time scales. In spite of the fact that the full sequences of variables cover the same time period in the environment, the different layers of the models function at separate levels of abstraction. The higher-level operates on a coarser timescale, implying that numerous lower-level time steps occur in a single higher-level step. The egocentric model operates on a fine-grained time scale $\tau$ and is responsible for dynamic decisions and path integration. The allocentric model operates on a coarser time scale $t$, where a sequence of poses $p$ over a period of time $t$ updates a specific location place $z_T$. In this model, at any time t, the pose $p_t$ and place $z_T$ can give back the corresponding observation $o_t$. At the topmost layer, the temporal resolution is lowest, where a single tick of the clock corresponds to a distinct location $l$, associated with the allocentric model at that time. This is carried out without accounting for the intermediary time steps of the lower layers.

This hierarchical arrangement allows the agent to reason about its environment further ahead, both temporally and spatially. In temporal terms, planning one step at the highest level (such as aiming to change location) translates to planning over multiple steps at the lower levels, and this pattern continues throughout the hierarchy. In spatial terms, the environment is organised in levels of abstraction, becoming more detailed as one descends the hierarchy (for instance, from connections between rooms to details of individual rooms).

In the following, we discuss the details of the models at each layer of the hierarchy using a bottom-up approach.

### 3.4.1. Egocentric Model

The egocentric model learns its latent state through the joint probability of the agent's observations, actions, policies, belief states and its corresponding approximate posterior. It comprises a **transition model** for factoring in actions when transitioning between states, a **likelihood model** for generating pixel-based observations and estimating collision probability based on the state, and a **posterior model** for integrating past events into the present state.

$$P(\tilde{o}, \tilde{c}, \tilde{s}, \tilde{a}) = P(s_0) \prod_{\tau=1}^{T} P(s_\tau | s_{\tau-1}, a_{\tau-1}) \prod_{\tau=1}^{T} P(o_\tau, c_\tau | s_\tau)$$

$$Q(\tilde{s}|\tilde{o}, \tilde{c}, \tilde{a}) = Q(s_0|o_0) \prod_{\tau=1}^{T} Q(s_\tau | s_{\tau-1}, a_{\tau-1}, o_\tau)$$

(8)

The egocentric model continuously updates its beliefs about the state (*s*) by incorporating the previous action (*a*) and the most recent visual observation (*o*) from the environment [37]. This belief correction process is described in Equation (8) and presented in Figure 2.

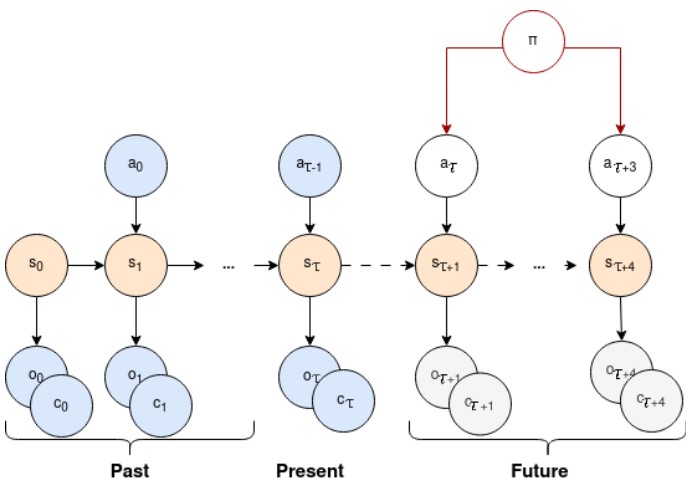

**Figure 2.** Generative model for the egocentric level: POMDP depicting the model transition from past and present (up to timestep $\tau$) to future (from timestep $\tau + 1$). A state $s_\tau$ is determined by the corresponding observation $o_\tau$ and influenced by the previous state $s_{\tau-1}$ and action $a_{\tau-1}$, generating the supplementary collision observation $c_\tau$. The action, as well as both observations, are assumed observable, indicated by the blue colour. In the future, the actions are defined by a policy $\pi$ influencing the new states in orange and new predictions in grey.

The incorporation of consecutive states forms the short-term memory of the model. It acquires an inherent comprehension of the dynamics of the environment through a process of trial and error, interacting with the environmental frontiers (e.g., walls). This learning is accompanied by the notion of action and consequences introduced by active inference. The observations of the model are visual observations (*o*) and dynamic collisions (*c*) in the environment.

The egocentric model serves as the lowest level of the overall model and is responsible for predicting the dynamic do-ability of policies. It discards any sequence of actions that is deemed impossible based on its understanding of the environment. Additionally, the egocentric model plays a crucial role in facilitating curiosity-driven exploration by making short-term predictions when the agent is uncertain about the beliefs of the allocentric model.

### 3.4.2. Allocentric Model

The allocentric model is responsible for generating environment states that describe the surroundings of the agent. It relies on generative query networks (GQN) [50,51]. To

form a conception of the agent's environment, its internal belief about the world is updated through interactions with the world, resulting in places (latent state $z$) structured upon positions ($p$) and corresponding observations ($o$) [50,51] as can be seen in Figure 3. The corresponding joint probability distribution $P(z, \tilde{o}, \tilde{p})$ defines, respectively, the agent's belief state, observations, poses, and the approximate posterior of this allocentric model:

$$P'(z, \tilde{o}, \tilde{p}) = P(z) \prod_{t=1}^{T} P(o_t | p_t, z) P(p_t)$$

$$Q'(z | \tilde{o}, \tilde{p}) = \prod_{t=0}^{T} Q(z | o_t, p_t) \tag{9}$$

This model, therefore, condenses chunks of information into a concise description of the environment. In this paper, we call one of these chunks a place, but it could also represent a context, as defined by Neacsu et al. [39]. In order to correctly condense information into the appropriate place, sequences of states at the lower level are separated using an event boundary based on the prediction error [52,53]. Each formed place (state $z$) represents a static structure of the environment. A dynamic environment results in new places being generated. The process of updating or generating a new place involves evaluating the agent's estimated global position within the cognitive map. This assessment results in closing the loop if the place is recognised or creating a new belief if it is not.

Each new place has its own local reference frame, created with a believed pose as the origin.

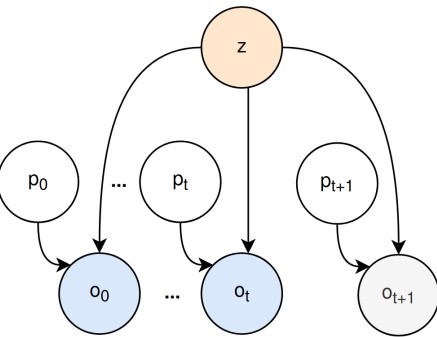

**Figure 3.** Generative model for the allocentric level as a Bayesian network. One place is considered and described by a latent variable $z$. The observations $o_t$ depend on both the place described by $z$ and the agent's position $p_t$. From 0 to t, the positions are visited and are used to infer a belief over the joint distribution. The future viewpoint $p_{t+1}$ has not been visited or observed yet. The observed variables are shown in blue, while the inferred variables are shown in white, and the predictions are presented in grey.

### 3.4.3. Cognitive Map

The cognitive map is responsible for memorising places and matching them with their relative positions in global space. It does this by creating nodes that we call experience or location. The creation of several experiences generates a metric-topological map of the environment, allowing the system to integrate the notion of distance and connections between locations.

A continuous attractor network (CAN) is employed to handle motion integration. This network processes successive actions across time steps, allowing the estimation of the agent's translation and rotation within a 3D grid [49]. The CAN's architecture, featuring interconnected units with both excitatory and inhibitory connections, emulates the behaviour observed in navigation neurons known as grid cells, found in various mammals [54], internally measuring the expected difference in the robot's pose (i.e., its coordinates x, y, and relative rotation over the z-axis). The CAN wraps around its edges, accommodating traversing spaces larger than the number of grid cells. The activation value of each grid cell

represents the model's belief in the robot's relative pose, and multiple active cells indicate varying beliefs over multiple hypotheses. The most highly activated cell represents the current most likely pose. Motion and proprioceptive translation modify cell activity, while view-cell linkage modifies activity when a place's latent state ($z$) differs significantly from others. This is determined through a cosine similarity score.

When a representation of an experience is stimulated, it adds an activation to the CAN at the stored pose estimate [41,55]. Each new combination of position and place ($z$) generated by the allocentric model develops a new experience in the cognitive map that is represented by nodes in a topological graph. Such a node integrates the view cell (place), the position, and the pose cell of the visited location [40]. Each place reference frame is mapped in the cognitive map global reference frame by remembering the local pose origin of the place reference frame and associating it with the location's global position. When the agent starts moving for the first time, the global frame is created with this first motion as the origin of the global reference frame.

When navigating, context is considered for closing loops. Meaning that when the current belief aligns with a past experience's place, the corresponding view cell activates. However, to resolve potential aliasing, the agent also considers its global position. If the position is determined to be too far from past experiences (based on a set threshold), a new place is created. This new place will adapt to new visual input without affecting the existing view cell associated with the past experience.

### 3.5. Navigation

The model is trained to learn the structure of the environment and should, therefore, be able to accomplish a variety of navigation tasks, regulated through active inference. Therefore, the agent can realise the following navigation tasks without needing any additional training.

**Exploration.** The agent is able to explore an environment by evaluating the surprise it can obtain from predicted paths.

**Reaching goals.** The agent can be given an observation as a preference and try to recall any past location matching this observation and plan the optimal path toward it or search for it.

To find a suitable navigation policy, we need to evaluate a range of policies, each considering a number of actions. To this end, we define a look-ahead parameter, defining the number of future actions when evaluating the candidate policy. Considering each possible action at each position is intractable with increasing look-ahead values, we limit the search to straight-line policies, as shown in Figure 4.

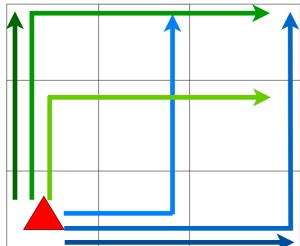

**Figure 4.** Illustration depicting L-shaped paths encompassing the upper right quadrant of an area surrounding the agent. The chosen look-ahead distance in this scenario is 2.

To establish those effective policies, we imagine a square perimeter around the agent with a width equal to the desired look-ahead. This square boundary is subsequently divided into segments, each regarded as distinct objectives. Our coverage approach involves crafting L-shaped paths originating from the agent's position and extending towards these segmented goals. By incrementally elongating the vector initiating from the agent, we ensure thorough area coverage. This strategy results in every position within the square area being approached from two divergent directions, as illustrated in Figure 4, within a

quarter of the square area. This methodology allows us to employ extended look-ahead distances without risking intractable calculations.

Once those policies are generated, the egocentric model evaluates their plausibility and truncates any sequence of actions leading to a collision with a wall. Using those plausible policies, the agent's navigation is guided by active inference. When the agent holds a high level of confidence in its world belief, its actions are determined by the variable weights in the following equation, leading it to either explore or pursue a specific goal.

$$
\begin{aligned}
G(\pi, T, \tau) = {} & \underbrace{W_1 \cdot \mathbb{E}_{Q'(o_\tau, z_T | \pi)} [\ln(Q'(z_T | \pi)) - \ln(Q'(z_T | o_\tau, \pi))]}_{\text{Allocentric Exploration}} \\
& + \underbrace{W_3 \cdot \mathbb{E}_{Q'(o_\tau | \pi)} [\ln(P'(o_\tau | g))]}_{\text{Allocentric Preference seeking}} \\
& + \underbrace{W_2 \cdot \mathbb{E}_{Q(o_\tau, s_\tau | \pi)} [\ln(Q(s_\tau | \pi)) - \ln(Q(s_\tau | o_\tau, \pi))]}_{\text{Egocentric Exploration}} \\
& + \underbrace{W_4 \cdot \mathbb{E}_{Q(o_\tau | \pi)} [\ln(P(o_\tau | g))]}_{\text{Egocentric Preference seeking}}
\end{aligned}
\tag{10}
$$

With $Q'$ and $P'$ being the approximate posterior and prior of the allocentric model and $Q$ and $P$ being the approximate posterior and prior of the egocentric model. The weights $W_i$ in Equation (10) are treated as adaptive model parameters. However, instead of being trained, they are increased or reduced depending on the certitude of the policies to lead to an estimated output, considering preferences when pertinent. This effectively regulates the different parts of the expected free energy depending on the current situation. If we have defined a preferred observation $g$, it effectively drives the agent toward reaching such an observation. Both the egocentric and allocentric models are used to infer the presence of the objective, using the same log preference mechanism. The egocentric model corrects possibly wrong memories of the allocentric model on the goal position in the immediate vicinity by out-weighting—with $W_4$—the allocentric model predictions when there is a discrepancy between the two. Therefore, while the egocentric model is trusted to infer the objective in its immediate vicinity, the allocentric model is trusted to search this objective in memory through all the previously visited places, from the latest to the oldest. For long-term planning between several places, the model aims to reach the place containing this preferred observation using active inference over the places leading toward the goal. Concretely, it means that there is little uncertainty, and the EFE is dominated by the utility term; hence, we use a shortest path algorithm such as Dijkstra [56] to determine the quickest path considering the distance between places, the number of places to cross, and the probability of a connection between places, allowing for a more greedy or conservative approach depending on the weight we put on probable and improbable connections between places. In this work, the inference is set as conservative, and unconnected places are considered unlikely to lead toward the objective faster. The agent moves from place to place by setting position observations leading from one place to the next as sub-objective $C$ in Equation (10). It moves by searching for this preferred observation $g$ while considering the direction it is headed toward to generate appropriate policies.

In the absence of any preference, the agent does not prioritise any particular observation; thus, the weights ($W_3$ and $W_4$) associated with preference seeking in both models are zero, prompting the agent to engage in exploration instead.

During exploration, the agent focuses on maximising the predicted information gain based on the expected posterior. Since the agent considers having a clear understanding of the environment after characterising a place, the uncertainty in observations becomes less relevant. As with preference seeking, if the allocentric model fails to identify a relevant policy to explore new territories, the egocentric model encourages the agent to venture beyond its familiar surroundings. It is important to recall that a latent state $z$ describes

one place and does not encompass the whole environment. Once the model considers that a place does not explain the observations anymore, it resets its beliefs and forms a new place. To imagine passing from one place to another, the cognitive map considers the agent-predicted location to shift the place of reference, which results in unvisited locations being much more attractive, as they have highly unexpected predictions, in contrast with visited places. An example of each layer's predictive ability is shown in Section 4.3

When transitioning between places, the allocentric model's confidence in the current place may drop below a pre-defined mean squared error (MSE) threshold between the prediction and the next step observation. In general, a number of steps are needed to build up confidence in the place visited given the observations. During this phase, Equation (10) is not employed for navigation. Instead, our primary goal is to ascertain the most accurate representation of the environment. To achieve this, the agent formulates hypotheses, involving new and memorised places $z_n$ and poses $p_t$, which potentially account for the observed data. The model strives to acquire additional data to converge towards a single hypothesis, accurately determining its spatial position.

In order to ascertain the best actions for acquiring observations that aid in convergence, Equation (11) is applied to each probable hypothesis $n$.

$$G(\pi, n) = W \cdot \sum_{t>i} \underbrace{\mathbb{E}_{Q'(z_n, p_t | o_{0:i+t}, \pi)}[\ln(Q'(z_n, p_t | \pi) - \ln(Q'(z_n, p_t | o_{0:i+t}, \pi)))]}_{\text{information gain}}$$
$$- \underbrace{\mathbb{E}_{Q'(z_n, p_t | o_{0:i+t}, \pi)}\big[\ln(P(o_t))\big]}_{\text{expected utility}} \qquad (11)$$

Hypotheses are weighted based on their alignment with the egocentric model's predictions. A hypothesis gains weight if its predictions closely match the expected observations. If no hypothesis stands out, they are considered equally probable.

Whatever the situation we are in, the leading policy is then inferred through

$$P(\pi) = \sigma(-\gamma G(\pi)) \qquad (12)$$

This effectively casts the planning as an inference problem, and beliefs over policies are proportional to the expected free energy. $\gamma$ offers a useful balance as it enables the elimination of policies that are highly unlikely, improving the efficiency of planning while also being relatively conservative [46].

*3.6. Training*

In order to effectively train this hierarchical model, the two lower-level models are considered independent and trained in parallel. To optimise the two ego-allocentric neural network models, we first obtain a dataset of sequences of action–observation pairs by interacting with the environment. This can be obtained, for instance, using a random policy, A-star-like policies, or even by human demonstrations. In this paper, the model was trained on a mini-grid environment consisting of 9 squared rooms in a layout consisting of 3 rows and 3 columns. Each room is composed of a number of tiles going from $4 \times 4$ to $7 \times 7$ tiles. Each room is assigned a colour at random from a set of four: red, green, blue, and purple, and connects to adjacent rooms by aisles of fixed length randomly placed, separated by a closed door in the middle. In addition, white tiles may be present at random positions in the map. The agent could start a training sequence from any door (or near door) position. An example of a randomly generated training environment is given in Figure 5. When the agent faces a door, it automatically opens and closes once the agent is no longer facing it (either by passing through or turning away). This feature allows the agent to focus on its motion behaviour. The training was realised in 100 environments per room width going from 4 tiles to 7 tiles. The agent has a top view of the environment covering a window of 7 by 7 tiles, including its own occupied tile. It cannot see behind itself nor through walls or closed doors. The observation the agent interprets is an RGB pixel rendering of

shape $3 \times 56 \times 56$ (see Appendix A.4 Figure A3 for an illustration of an observation). The allocentric model is trained on 1000 sequences per room size (4 to 7 tiles), each sequence has a random length of between 15 and 40 observations in a room that is separated between learning the room structure and predicting the observations given the pose and learned place (posterior). The model is optimised through the loss

$$\mathcal{L} = \sum_{t=0}^{T} D_{KL}[Q'\phi(z|o_t, p_t)||\mathcal{N}(0,1)] + ||\hat{o}_t - o_t||^2 \tag{13}$$

The approximate posterior $Q'$ is modelled by the factorisation of the posteriors after each observation. The belief over $z$ can then be acquired by multiplying the posterior beliefs over $z$ for every observation. We train an encoder neural network with parameters $\phi$ to enable the determination of the posterior state $z$ based on a single observation and pose combination $(o_k, s_k)$. The likelihood is optimised using the MSE, which involves the real observation $o_k$ and the predicted observation $\hat{o}_k$ [51]. To determine a position, the previous position and the agent's action are used to infer the next position. Then, a random set of (position, observation) pairs is shuffled to form the predictions used to optimise the model.

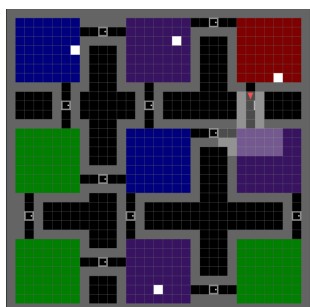

**Figure 5.** Example of a randomly generated map composed of 3 by 3 rooms of 7 tiles width each. Four white tiles are randomly disseminated in the environment. The red triangle represents the agent, and the highlighted squared area around the agent is its field of view.

The egocentric model is trained on 100 sequences of 400 steps per room size, and each full sequence is cut into sub-sequences of 20 steps. At each step, the model predicts what the observation should be and compares it to the real observation, improving its posterior and prior model parameters $\theta$ and $\phi$ through the loss function

$$\mathcal{L} = \sum_{t=1}^{T} D_{KL}[Q_\phi(s_t|s_{t-1}, a_{t-1}, o_t)||P_\theta(s_t|s_{t-1}, a_{t-1})] - log[P_\xi(o_t|s_t)] \tag{14}$$

This model is trained by minimising, in one part, the difference between the expected belief state given the last action and previous history and the estimated posterior state obtained given that action, observation, and updated history. In the second part, the difference between the reconstructed observation and the input observation is minimised [24], effectively optimising the likelihood parameters $\xi$. Both the egocentric and allocentric models are optimised using Adam [57].

The cognitive map, originally designed for navigation in mini-grid environments [25], can be re-scaled or adapted to different environments without the need for additional training.

## 4. Results

The objective of this paper is to propose a navigation model based on active inference theory in new similar-looking environments to which task requirements could be added. There is no definite benchmark to assess task-agnostic models; thus, our model is evaluated upon its particular ability to

- Imagine and reconstruct the environments the agent visited

- Create paths in complex environments
- Disambiguate visual aliases
- Use memory to navigate

In addition, the ability to explore an environment, as well as goal-reaching capabilities, are compared to competing approaches.

The model is tested in diverse mini-grid maze environments composed of connected rooms. Our agent is modelled to achieve autonomous navigation given only pixel-based observations.

To evaluate the effectiveness of the proposed model, a series of tests have been realised, each focusing on a specific aspect of the model. Those experiments range from evaluating the models composing the system to assessing its overall navigation performance. Even though the testing grounds are similar to the training set, all the tests were performed on environments the agent never saw during training.

### 4.1. Space Representation

The model's capacity to describe the observed place is critical to enable higher-level inferences. Therefore, the fewer observations it requires to achieve convergence to an accurate, or at the very least distinctive, representation of the environment, the more effectively it can recognise a place and navigate through it from various viewpoints. The model's rapid convergence is crucial, but it also needs to maintain adaptability, which involves the capability to incorporate new information about the place in its belief (such as discovering new corridors).

The following two figures demonstrate the place representation accuracy and convergence speed.

Figure 6 illustrates the inference process of place descriptions. Within approximately three steps, the main features of the environment are captured reasonably accurately based on the accumulated observations. Even when encountering a new aisle for the first time at step 11, the model is able to adapt and generate a well-imagined representation. Each observation corresponds to the red agent's clear field of view, as depicted in the agent position row (2nd row) of the figure (more details about the observations are in Appendix A.4).

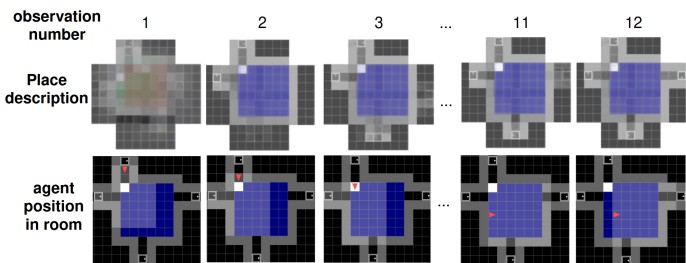

**Figure 6.** Evolution of the place representation in a room as new observations are provided by the moving agent (red triangle). The model is able to correctly reconstruct the structure of the room as observations are collected.

Figure 7 shows the agent consistently achieving a stable place description in about three observations in rooms having dimensions it has seen during training (rooms of 4 to 7 tiles width). Interestingly, the agent also exhibits the ability to accurately reconstruct larger rooms, even though it did not encounter such room dimensions during training. In particular, stable place descriptions for rooms composed of $8 \times 8$ tiles are attained in approximately five steps. This showcases the agent's allocentric model generalisation abilities beyond the limits of its training. The experiment was conducted over 125 runs in 25 environments with the agent tasked to predict observations from unvisited poses after each new motion. Figure 8 demonstrates the significance of the MSE value, used as a metric for this experiment, by displaying examples of predicted observations and their attributed

MSE values. In our experiments, to settle on a place to improve over successive steps, we set the MSE threshold to 0.5; thus, a prediction error above this threshold induces a reset of the current place and discards the previously accumulated observations and poses.

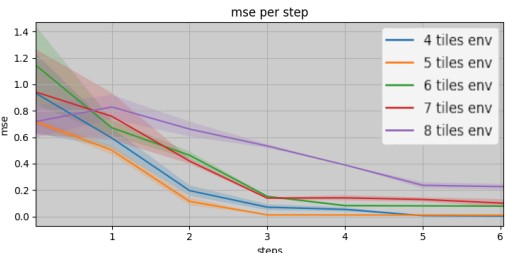

**Figure 7.** Prediction error of unvisited positions over 25 runs by room size, starting from step 0 where the models have no observation.

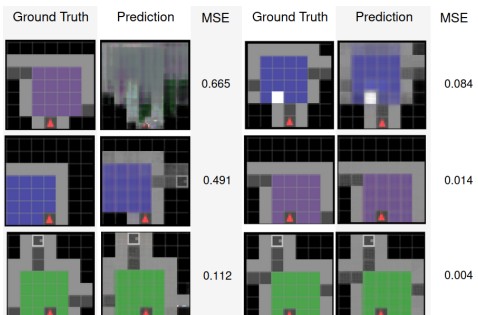

**Figure 8.** Observation ground truth, predicted observation, and MSE between observations.

The model demonstrates its ability to differentiate empty rooms based on their size, colour, and shape.

### 4.2. Navigation

Our navigation tests are focused on evaluating the model's ability to complete a well-defined task, such as forming a spatial map through exploration in an aliased environment. The agent is set to perform two tasks, environment exploration and goal reaching, without any additional training after learning the structure of familiar rooms.

**Baseline**. To establish a baseline for the navigation tasks, we compare our method against:

- C-BET [16], an RL algorithm combining model-based planning with uncertainty estimation for efficient exploration and decision-making.
- Random network distillation (RND) [58], integrates intrinsic curiosity-driven exploration to incentivise the agent's visitation of novel states, meant to foster a deeper understanding of the environment.
- Curiosity [59], leverages information gain as an intrinsic reward signal, encouraging the agent to explore areas of uncertainty and novelty.
- Count-based exploration [60] uses a counting mechanism to track state visitations, guiding the agent toward less explored regions.
- Dreamerv3 [5] represents an advanced iteration of world models for RL, offering the potential to enhance navigation by predicting and simulating future trajectories for improved decision-making.
- A-star algorithm (Oracle) [61] is a path planning algorithm to which the full layout of the environment and its starting position is given to plan the ideal path to take between two points.

Most of these models propose different RL-based exploration strategies for robotics navigation. All baselines have been trained and tested on the exact same environments as our model. For each model training detail, we refer to Appendix A.

The test environments consist of maze-like rooms that progressively increase in scale, ranging from 9 rooms up to 20 rooms, all with a width of 4 tiles.

### 4.2.1. Exploration Behaviour

We evaluate to what extent the hierarchical active inference model enables our agent to efficiently explore the environment. Without a preferred state leading the model toward an objective, the agent is purely driven by epistemic foraging, i.e., maximising information gain, effectively driving exploration [20].

Our evaluation involves comparing the performance of our model against various models such as C-BET, Count, Curiosity, RND models, and an Oracle. These models are tasked with exploring fully new environments with configurations ranging from 9 to 20 rooms. While the oracle possesses complete knowledge of the environment and its initial position, other models are only equipped with their top-down view observations (and, in the case of the RL models, extrinsic rewards). The RL models are encouraged to explore until they locate a predefined goal (white tile); however, the reward associated with the white tile is muted to encourage continued exploration. Notably, the DreamerV3 model faces challenges in effective exploration due to its reliance on visual observations of the white tile for reward extraction. Consequently, an adapted environment without the white tile or specific training would be necessary to employ DreamerV3 as an exploration-oriented agent in this study.

Across more than 30 runs by environment scale, our model demonstrates efficient exploration in terms of coverage and speed, comparable to C-BET and notably outperforming other RL models in all tested environments, as depicted in Figure 9, where we can see the percentage of area covered along steps in the environment. Moreover, the agent successfully achieves the desired level of exploration more frequently than any other model across all configurations, as demonstrated in Table 2. For an exploration attempt to be considered successful, the agents must observe a minimum of 90% of the observable environment. This criterion ensures that all rooms are observed at least once without imposing a penalty on the models for not capturing every single corner. Since the agents cannot see through walls (see Appendix A.4), entering a room may result in missing the adjacent wall corners, but these corners hold limited importance for the agent's objective. As an unlikely example, missing all the corner tiles of each room results in 9% of the environment not being observed (thus, no matter the scale of the environment). In this exploration task, the oracle stops exploring as soon as the exploration task is finished (exploring 90% of the maze), as can be seen in Figure 9, giving a good idea of what the ideal exploration should look like and the threshold they have to reach. However, to further analyse them, the other agents are requested to continue exploring upon completion of the task, thus, leading to over 90% maze coverage in the figures.

**Table 2.** The success rate of each model across all runs in each environment is defined as the percentage of runs where the exploration covers at least 90% of the environment.

| Success Rate (%) | Models | | | | |
|:---:|:---:|:---:|:---:|:---:|:---:|
| **Environment Configuration** | **Ours** | **C-BET** | **RND** | **Curiosity** | **Count** |
| $3 \times 3$ rooms | 93 | 81 | 16 | 32 | 13 |
| $3 \times 4$ rooms | 94 | 87 | 16 | 19 | 0 |
| $4 \times 4$ rooms | 91 | 81 | 26 | 16 | 0 |
| $4 \times 5$ rooms | 81 | 74 | 7 | 23 | 3 |

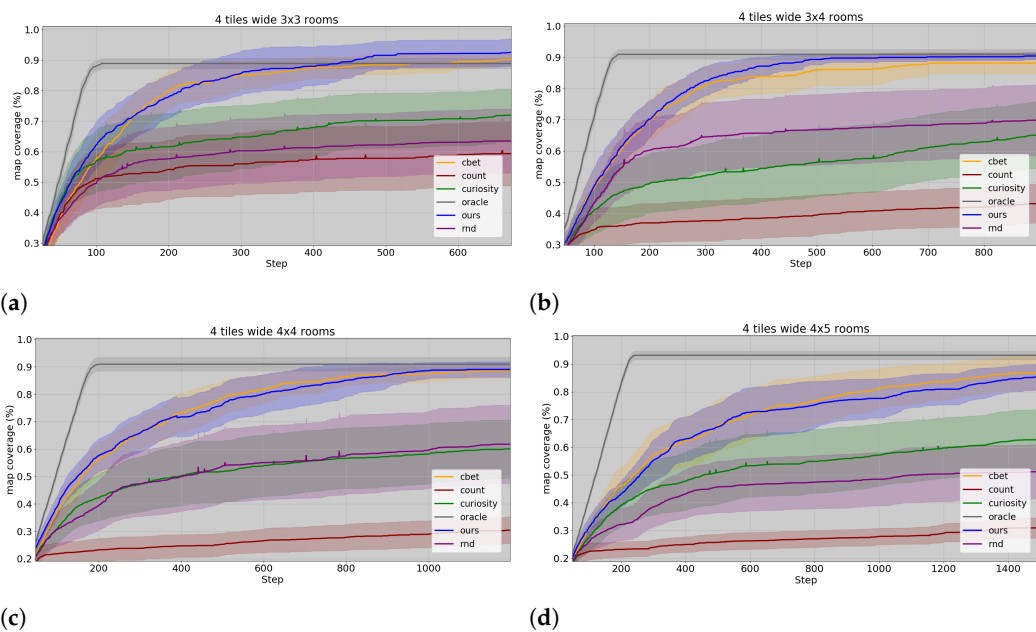

**Figure 9.** The average exploration coverage across all test instances (>30 runs) for each model computed for a given environment's scale. The oracle stops exploring as soon as the exploration task is finished (exploring 90% of the maze). (**a**) coverage as the exploration progress of all models in 3 by 3 room environments. (**b**) coverage as the exploration progress of all models in 3 by 4 room environments. (**c**) coverage as the exploration progress of all models in 4 by 4 room environments. (**d**) coverage as the exploration progress of all models in 4 by 5 room environments.

4.2.2. Preference Seeking Behaviour

To evaluate the exploitative behaviour of the models, we configure all the models mentioned in the baseline to navigate towards the single white tile within the environment. This is conducted across environments of increasing size, ranging from 9 to 20 rooms. Goal-directed behaviour is induced in our model by setting a preferred observation (i.e., the white tile), as typically occurs in active inference [1,20]. In our model, the preference for the white tile within the environment is not explicitly provided. Instead, the model is tasked with the objective of identifying a white tile based on its conceptual understanding of what the colour white represents. This approach enables the model to search for and recognise white tiles in its generated observations without direct access to the real observation in the tested environment. In the other RL models, an extrinsic and intrinsic reward is associated with this white tile in the environment, motivating the agents to explore until they reach this tile. The task is considered successful when the agent steps on the single white tile of the maze. A run is considered a failure if the agent has not reached the goal in under X number of steps, X depending on the world size. All models, except the oracle, start without knowing their position relative to the goal position in the environment. They need to explore until they find the objective. Figure 10 displays all the results by environment. The first column shows how much the models explore on average before reaching the goal and their success rate in diverse environments. Our model requires, on average, fewer steps than the other models to reach the goal, with the exception of the Count model. However, we can observe that Count also has the lowest success rate. The Count model often fails to reach the goal when it requires crossing several rooms. Overall our model reaches the white tile 89% of the time over all environments (see Table 3). Dreamerv3 is showing a poor performance because of over-fitting, not adapting well to new room configurations, and white tile placement it has never seen during training. This observation suggests that Dreamerv3 might require either a comparatively higher degree of human intervention or an extended dataset to effectively operate within our environment compared to other models.

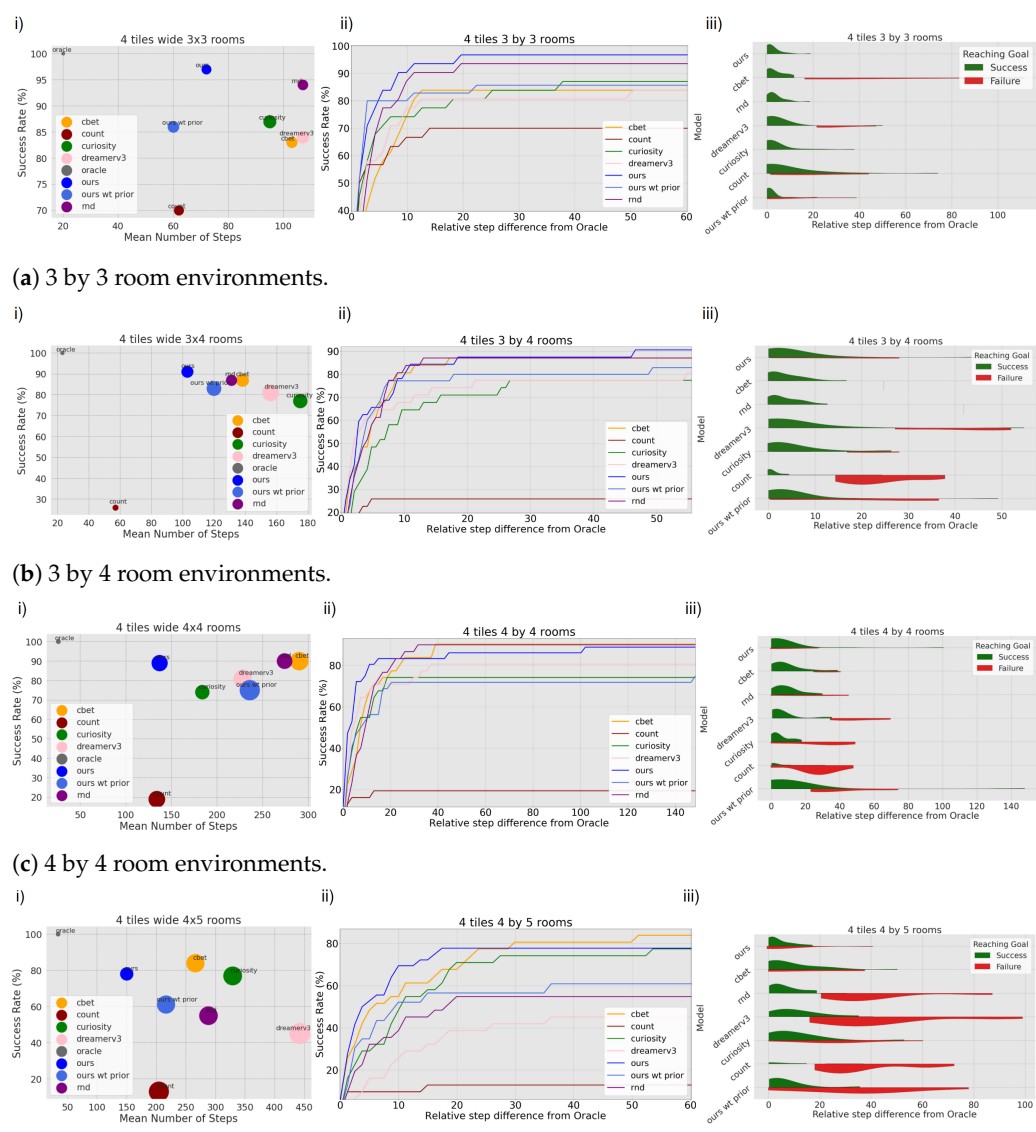

(**a**) 3 by 3 room environments.

(**b**) 3 by 4 room environments.

(**c**) 4 by 4 room environments.

(**d**) 4 by 5 room environments.

**Figure 10.** For environments ranging from (**a**) 3 × 3 to (**d**) 4 × 5 rooms, the results are presented in three graphs. The first column displays goal-reaching success rates and average steps. The second column illustrates the normalised deviations of each model's performance compared to the oracle, while the third column shows the distribution of success and failure based on normalised step deviations compared to the oracle.

The second column illustrates the proportion of goal attainment as the number of steps progressed relative to the oracle's optimal trajectory, normalised for comparison. Our model stands among the most efficient models to reach the goal rapidly, with 80% of the runs reaching the objective in less than ten times the steps taken by the oracle in most environments, except in the 4 by 5 room mazes.

Finally, the third column provides additional information about the proportion of success and failure according to the relative number of steps the oracle needs to reach the goal. From this plot, we can observe that most models are more likely to fail when the goal is far from the starting position. Our model, C-BET, Count, and Curiosity models show some failures at relative step 0 or before. This can be linked to the model returning an error due to excessive CPU consumption (in the case of C-BET, Count, and Curiosity) or by the agent believing a non-white tile to be white and sticking to it, terminating the task.

Our model's capacity to re-localise itself after positional disturbances allows us to conduct a supplementary experiment we call "ours wt prior". After permitting the model

to explore the environment, we teleport the agent back to its initial position and task it with seeking the goal. This experimental setup is exclusive to our model, which relies on a topological internal map for localisation. In contrast, other models in the baseline depend on sequential memory.

Intuitively, one might expect the model to achieve the goal more efficiently due to its internal map. Effectively, we observe that in 3 by 3 room mazes, 80% of successful runs reach the goal in less than three times the steps taken by the oracle, over 86% successful runs in total. However, the overall success rate is lower than the goal-seeking experiments without a prior. This discrepancy arises from various factors such as the quality of the map and navigation errors. The map generated during exploration can sometimes be imprecise, leading the agent to form erroneous assumptions about the location of the objective or guiding it along sub-optimal paths. When the model seeks a goal while having a prior understanding of the environment, it might pursue an incorrect objective approximately 35% of the time. In contrast, without any prior knowledge, the agent chases an erroneous objective around 29% of the time over all environments and runs. Additionally, in this condition, the agent seeks a path that surely leads to the objective and does not extrapolate over possible shortcuts. Therefore, if the shortest path leading to the goal goes through rooms that are not directly connected in the cognitive map, the path will not be optimal. Furthermore, the agent, guided by its priors, may not recognise a room while progressing towards the goal. This can result in the creation of a new experience that lacks proper connections to nearby rooms. Consequently, the agent might attempt to establish links with familiar rooms or backtrack in an effort to reach the room it did not initially recognise, wasting steps on those tasks. The agent's dependence on stochastic settings can lead to both failures and successes in similar situations, accounting for these varied outcomes. Despite that, the setting shows promise with a success rate comparable to other models.

**Table 3.** The success rate of each model across all environments and runs.

| Models | Oracle | Ours | C-BET | RND | Curiosity | Ours wt Prior | DreamerV3 | Count |
|---|---|---|---|---|---|---|---|---|
| success rate (%) | 100% | 89% | 86% | 81% | 79% | 76% | 72% | 31% |

### 4.3. Qualitative Assessments

Visual assessments of a specific environment are conducted to gain insights into the benefits of using a cognitive map for navigation. These assessments also involve evaluating the generated cognitive maps in comparison to the actual environment. Additionally, we compare exploration paths taken by various models to gain insights into their navigation strategies. Those few situations allow for a deeper insight into understanding the general behaviour of the model, including our own, shedding light on their navigation capabilities and the accuracy of our model's internal representation. A final consideration was given to the memory efficiency of each model during testing, and supplementary information on system requirements during training is available in Appendix A.1.

Our hierarchical model facilitates accurate predictions over extended timescales, over which the agent navigates between different rooms. In contrast, recurrent state space models commonly struggle when tasked to predict observations across room boundaries [50] or over long look-ahead windows [40]. Figure 11 illustrates the prediction capabilities of each layer over a prolonged imagined trajectory within a familiar environment. The figure showcases the predictions that each layer of the model creates as we project the imagination into the future, up to the point of transitioning to a new room and beyond. The last row demonstrates how the egocentric model gradually loses the spatial layout information over time, making it more suitable for short-term planning. The third row highlights the allocentric model's limitation to a single place in the environment, failing to recognise the subsequent room given current beliefs. Finally, in the second row, the cognitive map's imagined trajectory accounts for the agent's location and is capable of summoning the

appropriate place representation while estimating the agent's motion across space and time. The first row displays the ground truth trajectory, which aligns quite closely with the expectations of the cognitive map.

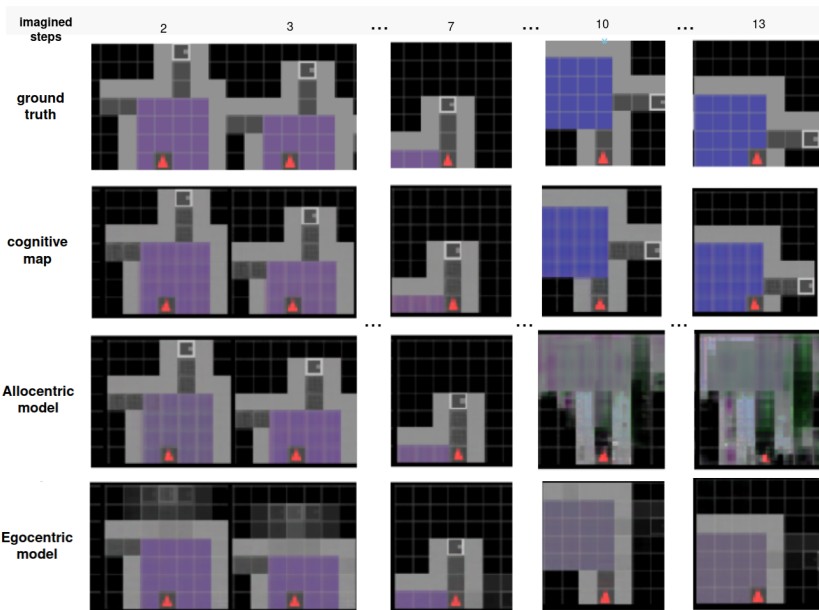

**Figure 11.** A trajectory leading toward a previously visited room is imagined by each model's layer. From bottom to top, the egocentric model, characterised by its short-term memory, gradually loses information as time progresses. This is evident from step 2 onward, where the front aisle is no longer present after the agent makes a few turns without visual input. In contrast, the allocentric model maintains the place description over time but encounters difficulty once it moves beyond the current place it occupies. The cognitive map, possessing knowledge of the connections between locations, accurately deduces the expected place behind the door, resulting in a prediction remarkably similar to the ground truth.

In order to navigate autonomously, an agent has to localise itself and correct its position given the visual information and its internal beliefs over the place. We performed navigation in a highly aliased mini-grid maze composed of four connected rooms having either the same colour, the same configuration, or the same colour and configuration but a single white tile of difference. Those four rooms are depicted in Figure 12A. The full Figure 12 illustrates the agent's exploration of the rooms and its ability to distinguish them without getting confused while entering rooms from a different aisle than previously.

Effectively, when the agent identifies a new place, it creates a new experience for it by considering its location. Figure 12B. displays each newly generated experience with a distinct ID and colour. To determine whether it enters a new place or comes back to a known one, it considers the probability of describing the current observations given each place, as can be seen in Figure 12C. The bars represent how many hypotheses are considered at each step, and the lines represent the probability of the place being a new one or a previously visited one. The colour of the lines corresponds to the experience's attributed colour in Figure 12B, blue lines being new unidentified places. Figure 12D. displays the internal representation of the places the agent uses. We can see that the rooms are accurately imagined, and even that a hesitation in an aisle position in Experience 1 is not enough to lose the agent.

In this context, the agent was able to successfully navigate and differentiate between rooms in a novel, highly aliased environment. The agent's ability to recognise previously visited rooms, even when entering from a new door, indicates its ability to maintain a spatial memory of the environment.

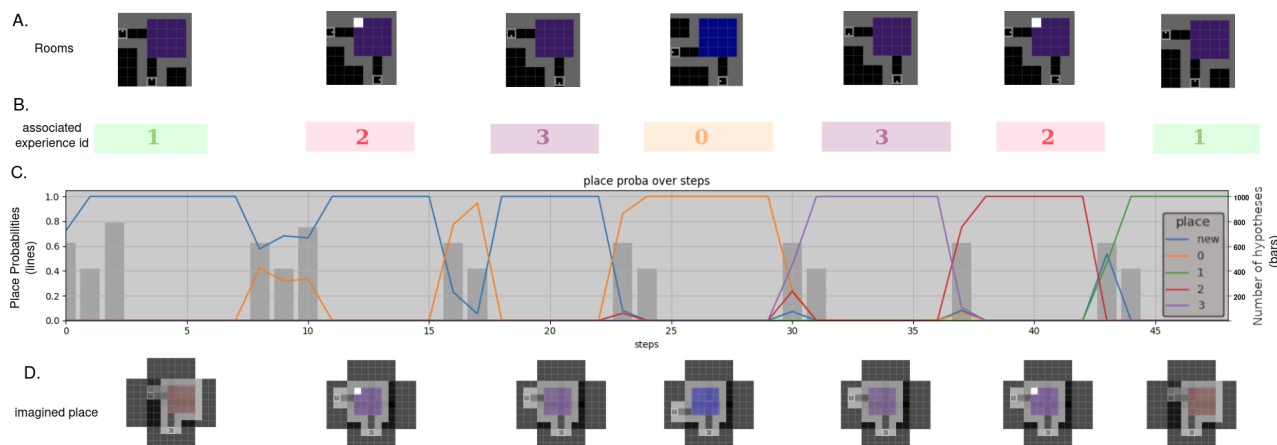

**Figure 12.** Navigation samples of the agent looping clockwise and anti-clockwise (thus, entering from a different door) in a new environment of 2 by 2 rooms over 142 steps. The clockwise navigation corresponds to a fully new exploration generating new places (see (**C**)), while the anti-clockwise loop leads through explored places. (**A**) a new world composed of 4 similar-looking rooms (colour or/and shapes), (**B**) the model associated with each room to a different experience ID corresponding to the place, (**C**) the probability of a new place being created (in blue, the most probable place among all possibilities) or an existing place being deemed the most probable to explain the environment. The grey bars represent how many new places are considered at once. The number of simultaneous hypotheses being considered can be read on the right part of the plot. (**D**) the imagined place generated for each experience id. We can see that Experience 1 is not fully accurate, yet it is enough to distinguish it from the other rooms given real observations.

Extending the experiment depicted in Figure 12, Figure 13 presents the complete trajectory's information gain according to the model. The graph exhibits a distinct pattern when exploring or exploiting, with the agent initially exploring the four rooms, as indicated by the fluctuating blue line, then retracing its path in identified rooms, indicated by colours relative to their ID. The information gain increases as the agent enters a new room, remains relatively steady while traversing within a place, and decreases during transitions between different places. When the agent retraces its steps at approximately step 100, the information gain becomes minimal, indicating that the agent has already gained knowledge about these locations. The info gain is higher or lower depending on how well it predicted the next observation, meaning the better its initial belief over the place, the lower the maximum accumulated information gain.

Throughout its exploration, the agent's curiosity plays a pivotal role, highlighting the significance of information gain in directing the agent's exploration towards unvisited areas rather than revisiting familiar places.

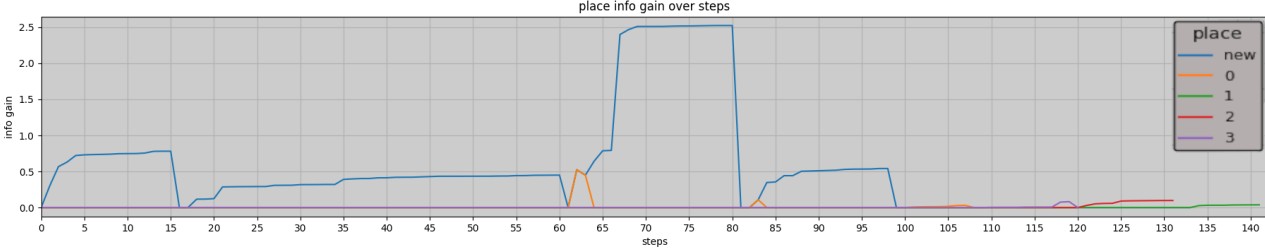

**Figure 13.** Information gain for each visited place. The blue curve corresponds to a new place being visited, while the coloured curves correspond to previously visited places, as presented in Figure 12. The first 100 steps correspond to the exploration of the agent of 4 different rooms, while the rest of the navigation corresponds to the re-visiting of those places. The information gain in a previously visited place is much lower than in a new room.

Figure 14 provides a direct comparison between the accuracy of the cognitive map's room reconstruction and the corresponding physical environment. This comparison reveals that the estimated map closely aligns with the actual map, with only minor discrepancies observed in some blurry passageways and a slight misplacement of the aisle in the bottom right room. This shows how important global position estimation is as the cognitive map uses the believed location to distinguish between two similar-looking rooms (purple rooms in the second column or blue rooms in the third column). This alignment between real and imagined maps underscores the fidelity of our model's internal representation in capturing the structural layout of the environment.

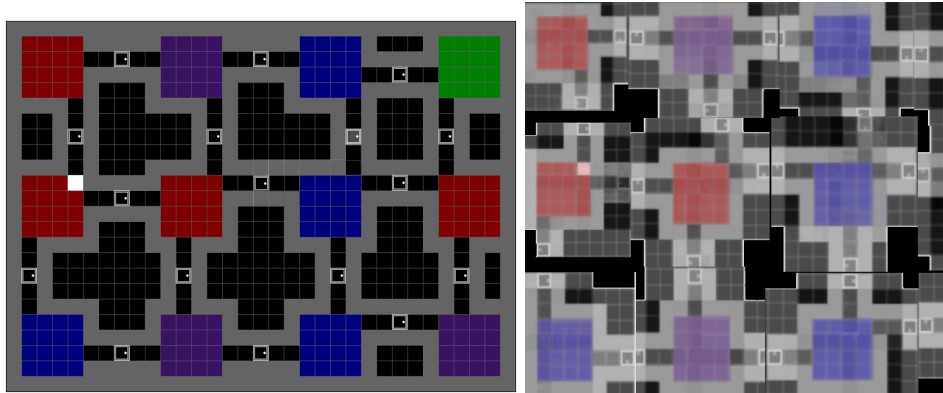

(**a**) Ground truth map of an environment.        (**b**) Reconstruction by the hierarchical model.

**Figure 14.** (**a**) displays the real map, while (**b**) is a composition of a cognitive map's room representations.

A correct internal mapping and layout structure definition allows our model to exhibit sensible decision-making when it comes to exploring the environment. Figure 15 presents an illustrative example of path generation for each exploration model in the same environment. The paths are represented by consecutive discrete steps from one tile to the next, with the progression from black (initial steps) to white (final steps). The oracle Figure 15a shows the most ideal path to observe 95% of the environment. Although lacking initial knowledge of the overall environment layout, our model demonstrates intriguing behaviour, as evidenced in Figure 15b. It exhibits a looping pattern, passing from the third to the first room. Upon realising the familiarity of the first room, the model subsequently alters its course to return to the third room and then explore the fourth room instead. It results in a complete exploration (100% of the tiles observed) in 212 steps, 151 steps less than C-BET Figure 15c. The Count model displays its inability to intelligently use doors to reach new rooms, over-exploring the same environment again and again. Its inefficiency probably comes from the observations being very aliased.

Our study demonstrates the capabilities of our agent to identify rooms rapidly and navigate to new places and back while resolving aliases and recognising previously visited environments, even when entering from a new location.

Finally, we observed computational constraints during testing. All the RL models presented in this study show a direct correlation between the number of steps in the environment and memory usage. Those RL methods use memory buffers to navigate efficiently. This often results in failure if the allowed memory capacity is insufficient to realise the task at hand. In contrast, our approach offers a more efficient solution, requiring less than 1 Gbyte of memory space, as well as avoiding scalability issues with respect to environment dimensions. Table 4 displays the peak memory requirements for each model across all tasks, with each run allocating a maximum of 1500 steps to complete the task.

Independently of those results, it seems relevant to remark that all the evaluated systems are computationally slow. The RL models become increasingly slower as the number of steps grows, attributable to the memory buffer. Our method experiences delays between

steps due to hypotheses calculations and policy evaluations, which are not parallelised and can scale significantly based on the space dimensions and look-ahead parameters.

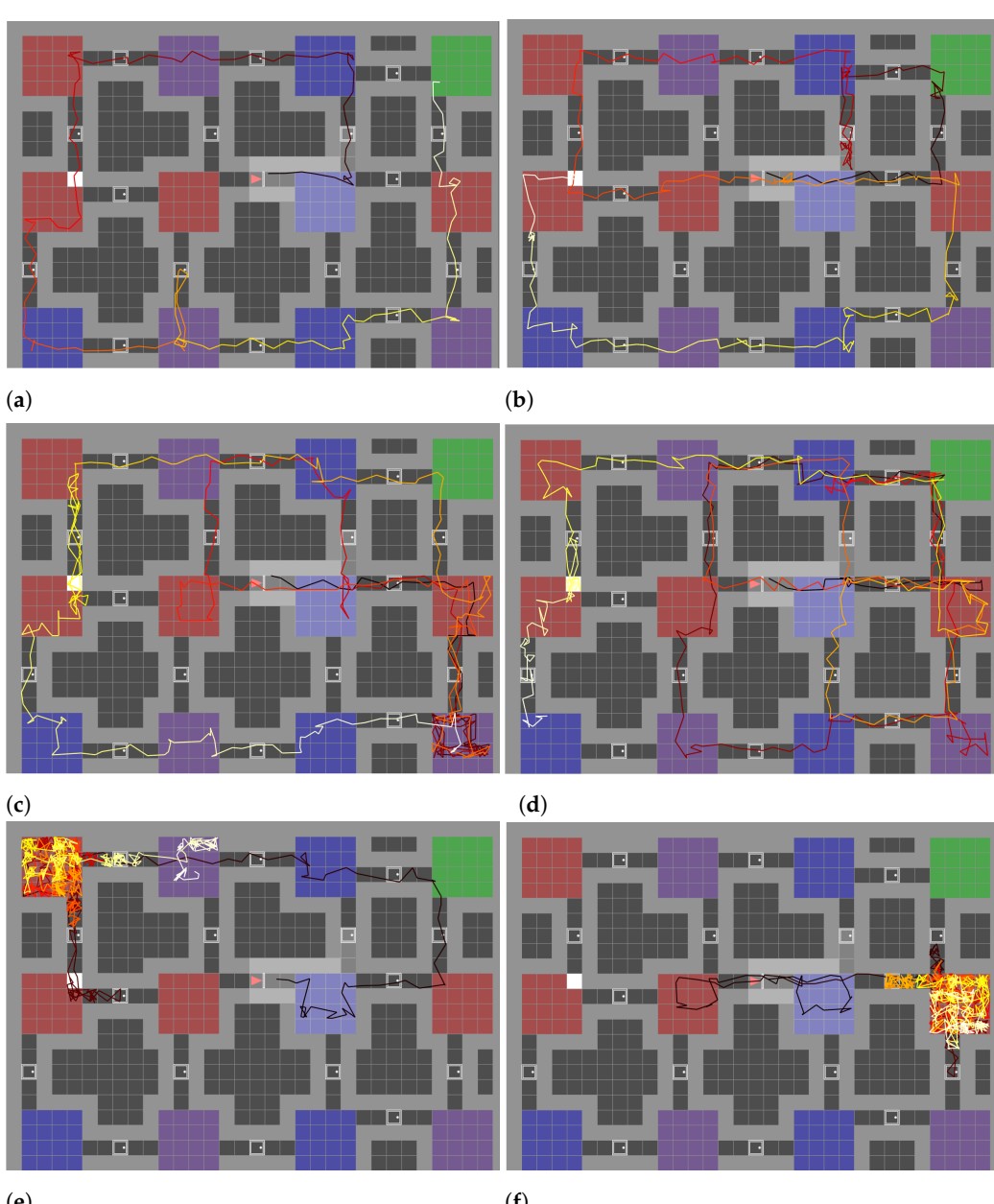

**Figure 15.** Paths taken by each model during an exploration run in the same 3 by 4 room environment. (**a**) The oracle path observed 95% (561 tiles) of the maze in 145 steps. (**b**) Our model path observed 100% of the environment in 212 steps (585 tiles). (**c**) The C-BET model path observed 100% of the environment in 363 steps (585 seen tiles). (**d**) The Curiosity model path observed 100% of the environment in 400 steps (585 seen tiles). (**e**) The RND model path observed 62% of the environment in 900 steps (364 seen tiles). (**f**) The Count model path observed 40% of the environment in 900 steps (235 seen tiles).

**Table 4.** Every model exhibits distinct system requirements. The following table highlights the most demanding criteria necessary for achieving successful exploration or goal seeking across the 4 by 5 room environments' configuration.

| Model | n° CPU | n° GPU | Used Memory (G) |
|---|---|---|---|
| Ours | 2 | 0 | 1 |
| Dreamerv3 | 2 | 1 | 28 |
| C-BET | 2 | 0 | 12 |
| RND | 2 | 0 | 9 |
| Curiosity | 2 | 0 | 11 |
| Count | 2 | 0 | 8 |

## 5. Discussion

We conclude with a comprehensive assessment of the proposed hierarchical active inference model for autonomous navigation, considering its strengths and limitations. We outline the key contributions of our work and discuss potential future works.

**Hierarchical active inference model**. Our proposal introduces a three-layered hierarchical active inference model:

- The cognitive map provides a unified spatial representation and memorises location characteristics.
- The allocentric model creates discrete spatial representations.
- The egocentric model assesses policy plausibility, considering dynamic limitations.

These layers collaborate at different time scales: the high level oversees the whole environment through locations, the allocentric model refines place representations as it changes position, and the egocentric model imagines action consequences.

**Low computational demands**. Our hierarchical active inference model has low computational demands, regardless of the environment's scale. This efficiency is particularly valuable as environments scale up, making our approach a potential solution for real-world applications.

**Scalability**. Our model efficiently learns spatial layouts and their connectivity. There exists the potential for our approach to adapt to novel scenarios by incorporating diverse environments into its learning process, thus, expanding allocentric representations. Furthermore, the possibility of introducing additional higher layers could facilitate greater abstraction, transitioning from room-level learning to broader structural insights.

**Task-agnostic**. The system does not require task-specific training, promoting adaptability in diverse navigational scenarios. It learns environmental structures and generalises to new scenarios, demonstrating its applicability to various objectives.

**Visual-based navigation**. Leveraging visual cues should enhance our model's real-world applicability.

**Aliasing resistant**. We show resistance to aliases, distinguishing between identical places and, thus, ideally supporting robust navigation.

While our approach offers several advantages, it is also important to acknowledge its limitations:

**Environment adaptation**. Our model requires adaptation to fully new environments for optimal performance. Training the allocentric model on room-specific data restricts navigation to familiar settings. To mitigate this and generalise to arbitrary environments, we could consider splitting the data by unsupervised clustering [62] or by using the model's prediction error to chunk the data into separate spaces [53].

**Recognition of changed environments**. Our proposal might struggle to detect environmental changes like altered tile colours, although this may not significantly impact

navigation performance as a new place will replace or be added with the previous one in the cognitive map. This remains an area for improvement.

In light of the features of our model listed above, as well as its limitations, our work offers a principled, biologically plausible approach to autonomous navigation. The integration of hierarchical active inference and world modelling enables our agent to navigate and explore an environment efficiently. Our model focuses on learning the structure of the environment and leveraging visual cues, aligning with the way animals navigate their surroundings, contributing to its real-world applicability.

Our experimental evaluation in mini-grid room maze environments showcases the effectiveness of our method in exploration and goal-related tasks. When compared to other reinforcement learning (RL) models such as C-Bet [16], Count [60], Curiosity [59], RND [58], and DreamerV3 [5], our hierarchical active inference world model consistently demonstrates competitive performance in exploration speed and coverage, as well as efficiency in reaching goals. Moreover, qualitative assessments show how accurate the cognitive map can be compared to the real environment and how the agent is able to differentiate aliased locations and use information gain to optimise navigation.

Our comprehensive assessment, both quantitative and qualitative, underscores the adaptability and resilience of our approach. As we move forward, there are several avenues for future research. The model's adaptation to new environments could be optimised, and methods for handling changes in familiar environments can be explored further. Additionally, exploration and goal-seeking tasks could be improved by adding a layer of comprehension to our cognitive map by integrating possible unexplored rooms when planning, in the form of potential places to visit [63]. Finally, the scalability and flexibility of our hierarchical structure could be extended to more complex, dynamic or realist scenarios such as Memory maze [64] or Habitat [65] to step toward real applications. This would require us to consider new challenges in place determination.

In conclusion, by combining principles of active inference and hierarchical learning, our hierarchical active inference model presents a preliminary solution, which promises to enhance autonomous agents' ability to navigate complex environments.

**Author Contributions:** Conceptualization, D.d.T. and T.V.; Methodology, D.d.T. and T.V.; Software, D.d.T.; Validation, D.d.T.; Writing—original draft, D.d.T.; Writing—review & editing, T.V.d.M., T.V. and B.D.; Supervision, T.V. and B.D.; Project administration, B.D. All authors have read and agreed to the published version of the manuscript.

**Funding:** This research received funding from the Flemish Government (AI Research Program) and the Interuniversity microelectronics centre (IMEC).

**Data Availability Statement:** Data are contained within the article.

**Conflicts of Interest:** Authors Toon Van de Maele and Tim Verbelen were employed by VERSES. The remaining authors declare that the research was conducted in the absence of any commercial or financial relationships that could be construed as a potential conflict of interest.

## Appendix A. Training Procedures

Each model necessitated specific considerations, which we outline below. We begin with an overview of the training system (see Table A1), followed by a description of the hyperparameters used for each model, highlighting any deviations from their source paper. Finally, we describe the observations used for each system.

### *Appendix A.1. System Requirements*

Each system required a different training time to reach the optimal behaviour. All the RL models were trained to optimise their policy, in contrast to ours, which moved randomly to learn the structure of the environment.

**Table A1.** Training characteristics for considered models. Insights into the training specifics of all models are provided, encompassing their respective training duration until reaching their finalised versions. Unfortunately, the information pertaining to RAM utilisation by the egocentric model is unavailable.

| Model | Training Time (h) | n° CPU | n° GPU | Used RAM (G) | Used Memory (G) | GPU Type |
|---|---|---|---|---|---|---|
| Ours egocentric | 32 | 4 | 1 | ? | 12 | GTX 980 |
| Ours allocentric | 95 | 2 | 1 | 2.5 | 20 | GTX 1080 |
| Dreamerv3 | 411 | 5 | 2 | 10 | 30 | GTX 1080 Ti |
| C-BET | 232 | 10 | 1 | 2.6 | 32 | GTX 980 |
| RND | 117 | 6 | 1 | 2.7 | 10 | GTX 980 |
| Curiosity | 90 | 6 | 1 | 3 | 10 | GTX 980 |
| Count | 141 | 6 | 1 | 2.7 | 11 | GTX 980 |

*Appendix A.2. Dataset*

Uniformity in training conditions was achieved by conducting training sessions for all models within identical environments, facilitated by the consistent application of a shared seed to generate these environments. The training environments consisted of mini-grid room mazes of 3 by 3 room configurations. These mazes were characterised by a range of room sizes, spanning from 4-tile width to 7-tile width, thereby constituting a total of 100 distinct rooms per room size.

*Appendix A.3. Hyper-Parameters*

All the benchmark models were trained using pre-set hyper-parameters, with C-BET, Count, Curiosity, and RND using the parameters described in Parisi et al. [16]. DreamerV3 uses the parameters proposed in Hafner et al. [5]; however, the behaviour of the model was modified from the original, setting an Exploring task behaviour and a Greedy exploration behaviour, as the original configuration was underperforming in our scenarios.

Our model was trained using the hyper-parameters in Table A2 for the allocentric model and Table A3 for the egocentric model.

**Table A2.** Allocentric model parameters.

| | Layer | Neurons/Filters | Stride |
|---|---|---|---|
| PositionalEncoder | Linear | 9 | |
| Posterior | Convolutional | 16 | 1 // (kernel:1) |
| | Convolutional | 32 | 2 |
| | Convolutional | 64 | 2 |
| | Convolutional | 128 | 2 |
| | Linear | $2\times 32$ | |
| Likelihood | Concatenation | | |
| | Linear | $256 \times 4 \times 4$ | |
| | Upsample | | |
| | Convolutional | 128 | 1 |
| | Upsample | | |
| | Convolutional | 64 | 1 |
| | Upsample | | |
| | Convolutional | 32 | 1 |
| | Upsample | | |
| | Convolutional | 3 | 1 |

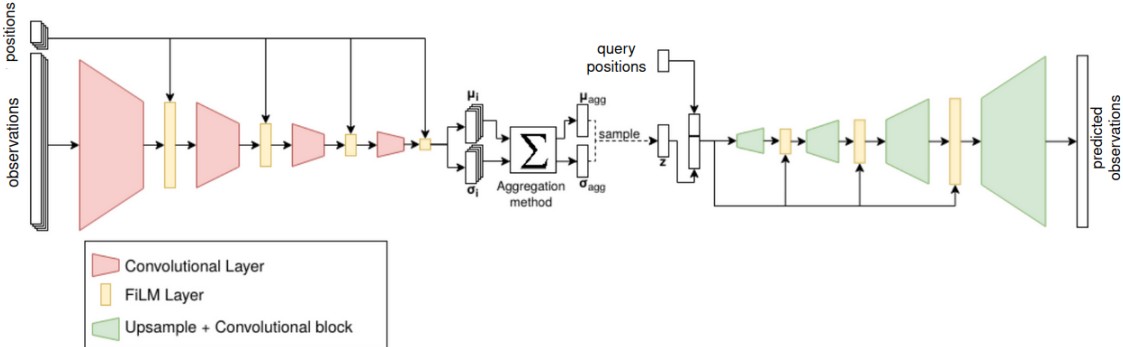

**Figure A1.** Schematic view of the generative model. The left part is the encoder that produces a latent distribution for every (observation, position) pair. This encoder consists of convolutional layers interleaved with FILM [66] layers that condition the positions. This transforms the intermediate representation to encompass the spatial information from the viewpoint. The latent distributions are combined to form an aggregated distribution over the latent space. A sampled vector is concatenated with the query position, from which the decoder generates a novel/predicted observation. The decoder mimics the encoder architecture, upsampling the image and processing it with convolutional layers, interfiled with a FILM layer that conditions the concatenated information vector.

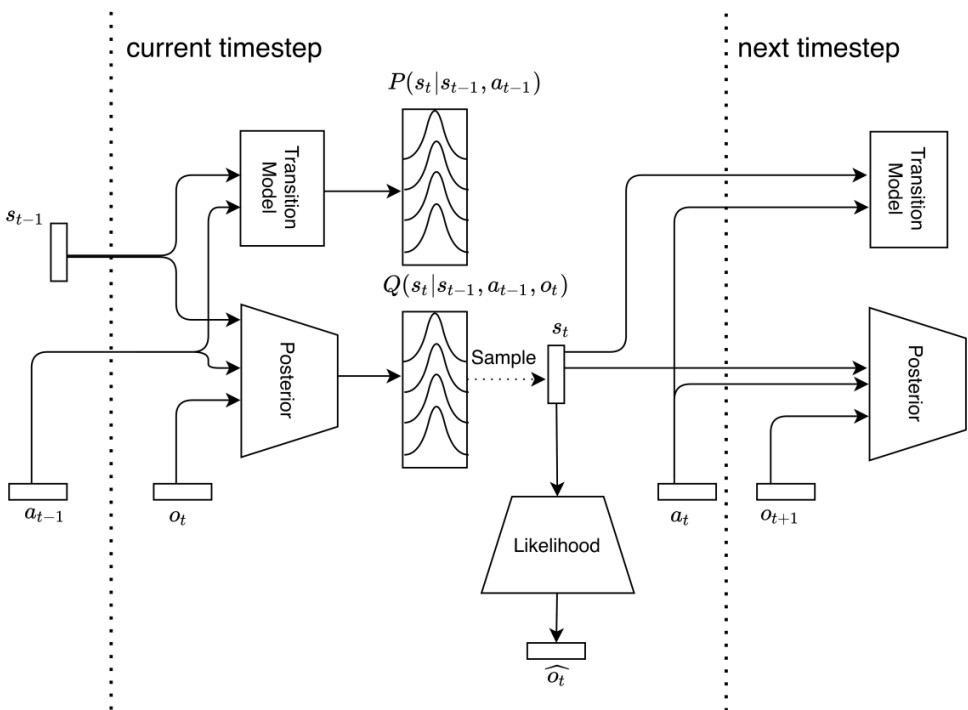

**Figure A2.** The generative model is parameterised by 3 neural networks. The transition model infers the prior probability of going from state $s_{t-1}$ to $s_t$ under action $a_{t-1}$. The posterior models the same transition while also incorporating the current observation $o_t$. Finally, the likelihood model decodes a state sample $s_t$ to a distribution over possible observations. These models are used recurrently, meaning they are reused every time-step to generate new estimates [37].

**Table A3.** Egocentric model parameters.

|  | Layer | Neurons/Filters | Stride |
|---|---|---|---|
| Prior | Concatenation | | |
|  | LSTM | 256 | |
|  | Linear | $2 \times 32$ | |
| Posterior | Convolutional | 8 | 2 |
|  | Convolutional | 16 | 2 |
|  | Convolutional | 32 | 2 |
|  | Concatenation | | |
|  | Linear | 256 | |
|  | Linear | 64 | |
| Image_Likelihood | Linear | 256 | |
|  | Linear | $32 \times 7 \times 7$ | |
|  | Upsample | | |
|  | Convolutional | 16 | 1 |
|  | Upsample | | |
|  | Convolutional | 8 | 1 |
|  | Upsample | | |
|  | Convolutional | 3 | 1 |
| Collision_Likelihood | Linear | 16 | |
|  | Linear | 8 | |
|  | Linear | 1 | |

*Appendix A.4. Model Observations*

All models use the agent's top-down vision of the agent, consisting of 7 by 7 tiles with the agent placed at the bottom centre of the image, as shown in Figure A3. Our model and DreamerV3 use an RGB view of the environment, while C-BET, Count, Curiosity, and RND use a one-hot encoded view of the environment, as well as an extrinsic reward when the agent passes over the single white tile in the environment. We can point out that the agent cannot see through walls in an RGB image. We can see in Figure A3a the environment and the agent's field of view represented by lighter colours. Figure A3b shows the actual observation seen by the agent.

The number of actions C-BET could take was greatly reduced compared to the original work, limiting it to actions such as forward, left, right, and standby. This was realised to maximise the similarity with our model and avoid a possible difference in performance due to an extended action space.

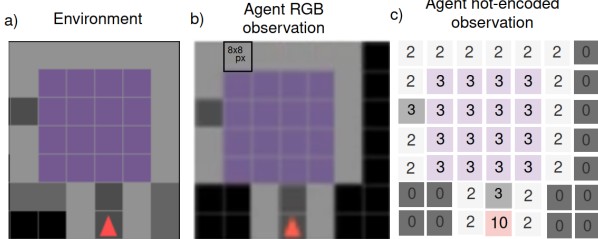

**Figure A3.** (**a**) cropped top-down view of the environment, (**b**) the RGB view of the agent. Each tile of the environment is composed of 8 by 8 pixels, generating a 56 by 56 total image. (**c**) The equivalent one-hot encoded view as a matrix. The numbers and colours are only relevant for the example.

All RL models had a sparse reward system, with an extrinsic reward generated only when passing on the white tile placed in the environment. Our model does not require rewards, and the goal we desire to set during the testing could be any kind of observation.

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
