# Peer review of "Spatial and Temporal Hierarchy for Autonomous Navigation Using Active Inference in Minigrid Environment"

_entropy, doi:10.3390/e26010083_

Round 1

Reviewer 1 Report

Comments and Suggestions for Authors

I would like to congratulate the authors on an excellent and innovative submission that makes a unique and meaningful contribution to the active inference literature by shining light on its applicability to autonomous navigation. I am happy to recommend this for publication in its current form (pending the correction of minor typographical and grammatical errors - please see below).

Otherwise, I do have one minor point - on p. 2, lines 41-45, the authors' initial definition of active inference is quite vague - perhaps they can briefly elaborate on this framework for the benefit of the unfamiliar reader?

Comments on the Quality of English Language

Throughout, there were a number of grammatical and typographical errors, particularly incomplete sentences. I would suggest another edit to correct these minor errors.

Author Response

Dear reviewer,

Thank you for your warm feedback and your comments.
We appreciate your suggestion to provide further details on the initial definition of active inference (AIF). In response, we have expanded upon the relevant sections to offer a more comprehensive understanding for unfamiliar readers. The specific modification can be found in attachment.
Regarding your concern about incomplete sentences, we have carefully reviewed and corrected the entire manuscript to ensure clarity and coherence. We hope that these revisions have addressed any potential issues with incomplete sentences and have improved the overall writing quality.

We are grateful for your time and effort in reviewing our work. If you have any additional suggestions or concerns, please do not hesitate to let us know.

Thank you once again for your valuable contributions.

Best regards

Reviewer 2 Report

Comments and Suggestions for Authors

Please see attached PDF for detailed comments. Almost all of my suggestions concern mainly fairly minor points in the language/presentation.

Comments on the Quality of English Language

Please see attached PDF for detailed comments. Almost all of my suggestions concern mainly fairly minor points in the language/presentation.

Author Response

Dear reviewer,
I sincerely appreciate your warm feedback and constructive comments on our manuscript.

We have carefully addressed all the concerns you raised, and corresponding modifications have been made to the paper. Your precise and thorough comments have been invaluable in improving our manuscript. To facilitate your review of the modifications, we have attached the revised manuscript with the main changes highlighted. 

I am genuinely grateful for your time and effort in writing this review of high quality. If you have any further comments or recommendations, please let us know.

Once again, thank you for your valuable contribution to the improvement of our paper.

Best regards

Round 2

Reviewer 2 Report

Comments and Suggestions for Authors

I am glad the authors found my feedback helpful. The concerns I raised have been addressed satisfactorily from my point of view, and the new additions are well-executed.